# Comparative Phytochemical, Colloidal, and Antioxidant Profiling of *Artemisia albida*, *Artemisia leucodes*, and *Artemisia scopaeformis*: Potentials for Cosmeceutical and Nutraceutical Applications

**DOI:** 10.3390/molecules30214165

**Published:** 2025-10-23

**Authors:** Janar Jenis, Ayaulym Minkayeva, Orynkul Yessimova, Saltanat Kumargaliyeva, Abdul Bari Shah, Thiruventhan Karunakaran, Kuzembekova Gulnur, Haji Akber Aisa, Aizhamal Baiseitova

**Affiliations:** 1Xinjiang Technical Institutes of Physics and Chemistry, Central Asian of Drug Discovery and Development, Chinese Academy of Sciences, Urumqi 830011, China; janarjenis@kaznu.kz; 2University of Chinese Academy of Sciences, Beijing 100039, China; 3Research Center for Medicinal Plants, Al-Farabi Kazakh National University, Almaty 050040, Kazakhstan; minkaeva.a@gmail.com (A.M.); abs.uom28@gmail.com (A.B.S.); 4Department of Analytical, Colloid Chemistry and Technology of Rare Elements, Faculty of Chemistry and Chemical Technology, Al-Farabi Kazakh National University, Almaty 050040, Kazakhstan; orinkul.esimova@kaznu.edu.kz (O.Y.); saltanat.kumargalieva@kaznu.edu.kz (S.K.); 5Research Institute for Natural Products & Technology, Almaty 050040, Kazakhstan; 6College of Pharmacy, Korea University, Sejong 30019, Republic of Korea; 7Interdisciplinary Major Program in Innovative Pharmaceutical Sciences, Korea University, Sejong 30019, Republic of Korea; 8Centre for Drug Research, Universiti Sains Malaysia, Penang 11800, Malaysia; thiruventhan@usm.my; 9Department of Microbiology, Virology and Immunology, Kazakh National Agrarian Research University, Almaty 050040, Kazakhstan; gulnur.kuzembekova@kaznaru.edu.kz; 10Division of Applied Life Science (BK21 Four), IALS, Gyeongsang National University, Jinju 52828, Republic of Korea

**Keywords:** *Artemisia* species, antioxidants, DNA oxidation protection, hydrolates, sun protection factor

## Abstract

*Artemisia albida*, *Artemisia leucodes*, and *Artemisia scopaeformis* were selected for this study based on their traditional medicinal uses and phytochemical profiles. *A. albida* exhibited the highest level of extractive substances (20.76%) and showed the greatest concentration of water-soluble polysaccharides (2.14%). Tannins, well known for their astringency and antioxidant activity, were most abundant in *A. scopaeformis* (2.81%) and *A. albida* (1.52%). The phenolics coumarins were concentrated in *A. scopaeformis* (6.49%) and *A. leucodes* (4.46%). Among the extracts, *A. leucodes* exhibited the strongest antioxidant activity (DPPH IC_50_ = 13.53 μM, FRAP = 52.02 μmol TE/g), the highest SPF (23.24), and the most effective DNA protection (91.4%). It indicated a high level of biological effectiveness, with an SPF comparable to natural UV-protection agents and DNA protection exceeding 90%, suggesting potential. Molecular docking confirmed binding of catechin and epicatechin to glutathione peroxidase. Colloidal analysis revealed that hydrolates obtained from *A. albida* and *A. leucodes* have significant surface activity, reducing water surface tension to 40–50 mJ/m^2^, whereas the hydrolate from *A. scopaeformis* had only a minor effect. Thus, *A. leucodes* is a strong candidate for multifunctional antioxidant, UV-protective, and skin-regenerating applications.

## 1. Introduction

Hydrolates are one of the most vital ingredients in cosmetics, due to their beneficial properties and natural origin, and currently their scope of application is very wide. They are a valuable byproduct of the essential oil extraction process obtained through steam distillation from plant parts [1]. Water vapor produced through the steam distillation process interacts with plant materials and extracts biologically active chemical components such as volatile oils, organic acids, flavonoids and vitamins [2,3]. In dermatology, hydrolates are valued for their naturalness, hypoallergenicity, and gentle effect on the skin, as they moisturize, soothe, and tone sensitive skin, making them popular in skincare and dermatology products [4]. Scientifically, hydrolates are valuable due to their natural and rich chemical compositions [5]. Study based on hydrolates allows researchers to develop safe and effective cosmetics, as well as explore potential biological properties for various medical and cosmetic applications [6]. The development of effective cosmetic products requires a detailed analysis of their colloidal and physicochemical properties, since most formulations are dispersed systems such as emulsions or suspensions. In this context, the colloidal–chemical properties of hydrolates obtained from *Artemisia* species native to Kazakhstan were determined as the purpose of this study. These plants were traditionally known for their immunomodulatory, antipyretic, tonic, warming, and laxative properties. They were employed in the treatment of conditions such as bronchial asthma, anemia, gastrointestinal disorders, diarrhea, chronic rheumatoid arthritis, and certain types of cancer. Additionally, numerous plant species were used for the prevention and treatment of dermatological conditions [7,8].

The genus *Artemisia* L., comprising more than 500 species, belongs to the family *Asteraceae* and is among the most widely distributed genera of medicinal plants worldwide. These species are found across various temperate regions, including Asia, Africa, Australia, and North America [9], with the highest species diversity occurring in Central and East Asia. For instance, China is home to 186 species, 82 of which are endemic; Russia records over 80 species; Kyrgyzstan has approximately 54 species, including 5 endemics; Uzbekistan contains 47 species; Turkmenistan hosts 33 species, with only 1 endemic; and Kazakhstan is home to 81 species, 19 of which are endemic [10]. Phytochemical studies have demonstrated that *Artemisia* species in Kazakhstan are rich in bioactive compounds, including sesquiterpene lactones, lignans, flavonoids, coumarins, polysaccharides, alkaloids, and various extractives, many of which exhibit significant therapeutic potential [11].

Three *Artemisia* L. species native to Kazakhstan (*Artemisia albida* Willd., *Artemisia leucodes* Schrenk., and *Artemisia scopaeformis* Ledeb) remain underexplored with respect to their therapeutic potential. *A. albida* is a semi-white medicinal plant distributed in the Altai, Tarbagatai, Zaysan, and Akmola regions of East Kazakhstan [9]. The first comprehensive phytochemical study of these species led to the identification of several sesquiterpene lactones, including austricin, matricarin, canin, and agrolide [12]. Additionally, anhydro-austricin was later isolated from *A. albida* [13]. This compound exhibits pharmacological properties such as angioprotective and antilipidemic activity, although it demonstrates minimal antimalarial efficacy [14]. *A. leucodes*, an annual or biennial herbaceous species with a whitish appearance, is predominantly found in the desert zones of Kazakhstan [15]. Phytochemical analysis of its aerial parts resulted in the isolation of five sesquiterpene lactones: anhydroaustricin, matricarin, leucomizin, grossmizin, and 5β(H)-austricin16. These compounds have been reported to possess various bioactivities, including antibacterial, angioprotective, hypolipidemic, and phagocytosis-stimulating effects [14,16]. Furthermore, the essential oil of *A. leucodes* is rich in biologically active components such as camphor, 1,8-cineole, and camphene, which contribute to its therapeutic potential [17]. The third species, *A. scopaeformis*, is an endemic plant native to Kazakhstan and remains largely unstudied. It grows in arid regions with clay and sandy soils, particularly in the Chu-Ili Mountains, Karatau, and Kyzylorda [9]. Its chemical composition was first investigated at the Research Center of Medical Plants in Almaty, Kazakhstan. The phytochemical profile of chloroform and hexane extracts was analyzed using Gas Chromatography–Mass Spectrometry (GC-MS). The hexane extract was found to contain four major constituents: methyleugenol; hexadecanoic acid, ethyl ester; butyl 4,7,10,13,16,19-docosahexaenoate; hexasiloxane, a1,1,3,3,5,5,7,7,9,9,11,11-dodecamethyl. Analysis of the chloroform extract revealed five additional compounds such as fluorene, 2,7-bis(1-hydroxyethyl); p-dimethylaminobenzylidene p-anisidine; 3-acetoxy-5-methyl-2-nitro-terephthalic acid, 4-isopropyl ester 1-methyl ester; 3,4-diacetyl-2-methyl-4H-thieno [3,2-b] pyrrole-5-carboxylic acid, methyl ester; 4H-1,2,4-triazole-3-thiol, 4-(2-fluorophenyl)-5-(1-methylethyl). These compounds are associated with a range of pharmacological activities, including antioxidant, antifungal, antimicrobial, anti-inflammatory, and anticancer properties [18].

Thus, a detailed chemical, colloidal, and biological investigation of these three *Artemisia* L. species was undertaken as the central objective for obtaining unique hydrolates derived from these medicinal plants. To achieve this objective, a comprehensive scientific study was conducted on the phytochemical composition, antioxidant potential, and colloidal properties of hydrolates obtained from three *Artemisia* L. species growing in different geographical regions of Kazakhstan. Qualitative and quantitative analyses of bioactive compounds were performed using high-performance liquid chromatography (HPLC), spectrophotometry, and titrimetry. Antioxidant activity was assessed through the DPPH and FRAP assays, while additional evaluations included sun protection factor (SPF), DNA protection against oxidative damage, and molecular docking to examine interactions between active compounds and biological targets. Moreover, the colloidal characteristics of the hydrolates were evaluated, including surface tension, pH, and membrane permeability, along with their emulsifying and foaming capacities. The results obtained are considered in the context of their practical application in the development of cosmetic and therapeutic formulations, such as creams and ointments.

## 2. Results and Discussion

### 2.1. The Quantitative and Qualitative Analysis

Before the quantitative identification, plant raw materials were assessed for moisture and ash content. Moisture content was relatively similar across all samples: *A. albida* (8.19%), *A. leucodes* (8.11%), and *A. scopaeformis* (8.08%), indicating good stability of the plant materials. Ash content varied slightly, with *A. leucodes* showing the highest value (7.99%), followed by *A. scopaeformis* (7.11%) and *A. albida* (6.78%). The higher ash content in *A. leucodes* suggests a richer mineral composition.

A comprehensive phytochemical analysis was carried out to quantify and compare key groups of bioactive compounds present in the aboveground parts of the three species. Based on the results obtained from the analysis, the composition of bioactive constituents such as flavonoids, alkaloids, coumarins, polysaccharides, extractive substances, tannins, and organic acids was identified and determined in all three *Artemisia* species (Table 1). In addition, moisture content and ash values were assessed as part of the physicochemical evaluation.

As shown in Table 1, extractive substances were present in relatively high amounts: *A. albida* (20.76%), *A. leucodes* (14.39%), *A. scopaeformis* (18.43%). *A. albida* showed the highest concentration of extractive substances (20.76%) and the most abundant with water-soluble polysaccharides (2.14%), which are known for their immunomodulatory and wound-healing effects. Tannins, which possess notable astringent and antioxidant properties, were found in the highest amounts in *A. scopaeformis* (2.81%) and *A. albida* (1.52%). Coumarins were especially concentrated in *A. scopaeformis* (6.49%) and *A. leucodes* (4.46%), suggesting significant anticoagulant and antimicrobial activities.

### 2.2. Mineral Composition

The ash content of three *Artemisia* L. species was analyzed for macro- and microelements using multi-element atomic emission spectral analysis, as presented in Table 2. The results of the analysis show that the predominant elements in all three species are calcium (Ca), potassium (K), and magnesium (Mg). In *A. albida*, the concentrations of Ca, K, and Mg were 2.87 mg/g, 3.60 mg/g, and 1.049 mg/g, respectively. In *A. leucodes*, the concentrations were 5.00 mg/g for Ca, 6.60 mg/g for K, and 1.49 mg/g for Mg. In *A. scopaeformis*, the concentrations were 6.61 mg/g for Ca, 10.91 mg/g for K, and 2.06 mg/g for Mg. These results indicate that *A. scopaeformis* is the richest source of essential macroelements among the three species studied. In particular, the high K and Ca content suggests potential benefits for skin hydration, barrier repair, and anti-inflammatory action, which are critical for dermatological applications. The presence of Mg, although lower, may also contribute to cellular regeneration and antioxidant defense. Overall, the mineral composition supports the potential use of these species in skin-supportive formulations. Ca, K, and Mg, found in the samples, are essential micronutrients that help maintain healthy skin. Ca promotes epidermal differentiation and barrier restoration, while K helps regulate cellular hydration and osmotic balance. Mg plays a key role in antioxidant protection, stabilizing the cellular redox status and protecting against oxidative stress. The synergistic action of these minerals, combined with polyphenolic compounds, enhances the overall dermal potential of the extracts, providing complementary mechanisms of protection, hydration, and skin regeneration.

### 2.3. Semi-Quantitative Determination of Polyphenolic Content by HPLC

To determine the presence of selected phenolic compounds, High-Performance Liquid Chromatography (HPLC) analysis was performed on *Artemisia* species. The polyphenolic profile was established using five standard compounds: gallic acid, catechin, epicatechin, naringin, and phlorizin. The analysis revealed the detection of these five compounds in the samples at varying concentrations. Table 3 provides the retention times (in minutes) and the validity test for determination of concentrations of the standard compounds. Compounds were selected due to their well-documented antioxidant and UV-protective activities, making them suitable markers for evaluating the photoprotective potential of *Artemisia* species.

Each of the three species were fractionated with ethyl acetate (EA) and butanol (BuOH), as they are known for the highest phenolic content, and the respective fractions were analyzed for their phenolic content to compare phenolic constituents. Various phenolic and flavonoid constituents were identified at 272 nm, indicating that the extracts were notably rich in these compounds.

The results of the HPLC analysis revealed significant variation in the phenolic composition across the different *Artemisia* species and solvents used, as summarized in Table 4 and Figure 1. The highest concentration of gallic acid was observed in *A. albida* in the butanol extract (3.41 ± 0.11 µg/1 g of dry mass of the plant), while *A. scopaeformis* and *A. leucodes* butanol extracts displayed lower concentrations of 2.49 ± 0.22 and 2.36 ± 0.10 µg/1 g dw, respectively. Catechin content was highest in *A. scopaeformis* in the butanol extract (271.21 ± 2.46 µg/1 g of dry mass of the plant), whereas *A. albida* and *A. leucodes* showed lower levels.

Epicatechin was most abundant in *A. scopaeformis* (BuOH—45.73 µg/g; EA—17.74 µg/g) and *A. leucodes* (BuOH—26.64 µg/g; EA—15.80 µg/g), while *A. albida* exhibited considerably lower concentrations (BuOH—3.72 µg/g; EA—2.40 µg/g). Naringin was the predominant compound in the ethyl acetate extracts of *A. scopaeformis* (386.73 µg/g) and *A. albida* (354.32 µg/g), with significantly lower concentrations observed in the butanol extracts (38.30 and 11.48 µg/g, respectively). Regarding phlorizin, the highest concentration was recorded in *A. albida* in the ethyl acetate extract (491.90 µg/g), while *A. leucodes* and *A. scopaeformis* EA showed peak values of 64.43 µg/g and 164.49 µg/g, respectively. Overall, *A. scopaeformis* exhibited the highest total flavonoid content, particularly when extracted with ethyl acetate. In contrast, butanol proved to be a more effective solvent for extracting catechin and epicatechin. The presence of naringin and phlorizin in the analyzed *Artemisia* extracts is consistent with previous findings on related species. For instance, *A. ludoviciana* was reported to contain flavanone-7-O-glycoside (naringin) as one of its main flavonoids, while *A. arborescens* and *A. inculta* from Crete were also shown to possess phlorizin among their secondary metabolites [19,20]. These observations suggest that both naringin and phlorizin may occur more broadly across the *Artemisia* genus, contributing to its characteristic antioxidant and protective properties.

### 2.4. Antioxidant Activity of Artemisia Species

Previous studies have established that *Artemisia* species, particularly *A. albida*, *A. leucodes*, and *A. scopaeformis*, are well known for their diverse bioactive compounds, including flavonoids, phenolic acids, and terpenoids. These compounds are associated with a range of bioactivities, such as antioxidant, anti-inflammatory, and UV-protective effects. Due to their ethnomedicinal use and the reported high antioxidant content, these species were selected as promising candidates for the evaluation of their antioxidant, photoprotective, and DNA-protective properties.

The antioxidant and protective effects of *A. albida*, *A. leucodes*, and *A. scopaeformis* were assessed using two different assays, DPPH radical scavenging activity (IC_50_) and ferric reducing antioxidant power (FRAP). The DPPH (2,2-diphenyl-1-picrylhydrazyl) assay is a commonly used method to evaluate the antioxidant activity of compounds. This assay is based on the reduction of the stable violet DPPH radical by antioxidants, which results in a decrease in absorbance at 517 nm, indicating the scavenging ability of the test sample.

Figure 2 illustrates the DPPH radical scavenging activity of *Artemisia* extracts at different concentrations. Among the extracts (Table 5), *A. leucodes* exhibited the highest antioxidant activity, with a DPPH IC_50_ value of 13.53 ± 0.71 μM, indicating strong free radical scavenging ability, comparable to the standard antioxidant Trolox (IC_50_ = 10.21 ± 1.06 μM). *A. albida* demonstrated moderate antioxidant activity with an IC_50_ value of 20.04 ± 1.27 μM, while *A. scopaeformis* showed the weakest radical scavenging ability, with an IC_50_ value of 28.87 ± 0.85 μM.

The FRAP assay measures antioxidant capacity by assessing the reduction of the Fe^3+^-TPTZ (iron(III)-tripyridyltriazine) complex to Fe^2+^-TPTZ in the presence of antioxidants. The formation of a blue-colored complex is detected at 593 nm, with higher absorbance indicating stronger reducing power. The results of the FRAP assay further corroborate this trend, with *A. leucodes* displaying the highest reducing power (52.02 ± 3.61 μmol TE/g), followed by *A. albida* (43.91 ± 2.31 μmol TE/g) and *A. scopaeformis* (32.37 ± 1.22 μmol TE/g). These findings suggest that *A. leucodes* possesses the most effective redox potential, which correlates well with its DPPH results, as they assess antioxidant capacity through electron transfer mechanisms, and the observed trends were consistent across both assays.

### 2.5. Sun Protection Factor (SPF) Effect of Artemisia Species

The SPF determination aimed to assess the UV absorption capacity of the extracts, which is important for potential sunscreen applications. The SPF values, which indicate the potential for UV protection, followed a similar trend, with *A. leucodes* exhibiting the highest SPF value (23.24 ± 0.71), suggesting promising photoprotective potential. It indicates a moderate to high level of UV protection, which is remarkable for a plant-based formulation. This value is comparable to several benchmark sunscreen agents such as octyl methoxycinnamate or titanium dioxide used at low concentrations. Such a result highlights the potential of the extract as a natural photoprotective ingredient with both antioxidant and UV-absorbing properties, suggesting its suitability for incorporation into multifunctional dermaceutical formulations. *A. albida* and *A. scopaeformis* had lower SPF values of 14.59 ± 1.03 and 9.93 ± 0.92, respectively, indicating relatively weaker UV absorption capacities. Figure 3 shows the UV absorbance of *Artemisia* extracts at concentrations of 0, 25, 50, 100, 200, 400, 800, 1600, 3200, 6400, 12,800, and 25,600 µg/mL in the range of 200 to 400 nm.

### 2.6. DNA Oxidation Protection of Artemisia Extracts

The DNA protection assay was performed to evaluate the ability of *Artemisia* extracts to prevent oxidative DNA damage, a critical aspect for assessing their potential to mitigate genetic damage and enhance cellular protection. In this assay, supercoiled circular DNA (scDNA) was converted into open circular DNA (ocDNA) and further into linear double-stranded DNA (lnDNA) by hydroxyl radicals generated through the Fenton reaction. The efficacy of the *Artemisia* extracts in preventing oxidative damage to plasmid DNA was assessed using a standard procedure [21].

All three extracts demonstrated significant protection against DNA damage, with *A. leucodes* providing the highest level of protection (91.4%), followed by *A. albida* (86.3%) and *A. scopaeformis* (82.1%). These results highlight the potential of these species in protecting against genetic damage induced by oxidative stress. Figure 4A shows the protective effects of the extracts on pBR322 plasmid DNA at a concentration of 100 μM. DNA derived from the pBR322 plasmid exhibited two distinct bands in the agarose gel. The faster-moving band corresponded to the native form of scDNA, while the slower-moving bands represented ocDNA. The most active extract, *A. leucodes* (Figure 4B), exhibited a dose-dependent protection against DNA damage, with concentrations ranging from 12.5 to 100 μg/mL.

Overall, *A. leucodes* exhibited the strongest antioxidant, UV-protective, and DNA damage-preventing properties (Table 5), positioning it as a promising candidate for further investigation in nutraceutical and cosmeceutical applications. A visible trend between the DPPH, FRAP, and SPF values suggests that the bioactive compounds in these extracts may contribute to multiple protective effects.

### 2.7. Molecular Docking Experiments

Finally, molecular docking analysis was conducted to elucidate the interactions of the primary phytochemicals with glutathione peroxidase (GPx), a critical enzyme in the skin’s antioxidant defense system. The five main compounds in the extracts—catechin, epicatechin, naringin, phlorizin, and gallic acid—were selected to investigate their potential binding interactions with GPx, an antioxidant enzyme that plays a vital role in protecting skin cells from damage caused by free radicals. All five compounds demonstrated potential binding to GPx via non-covalent interactions, including hydrogen bonding, aromatic hydrogen bonding, and π-interactions, as shown in Figure 5. The glide scores of −7.08 and −7.70 for catechin and epicatechin, respectively, suggest strong binding interactions, followed by naringin and phlorizin (both with glide scores of −6.52), and gallic acid, which exhibited a glide score of −5.64.

These results suggest that catechin and epicatechin exhibit the strongest affinity for GPx, potentially enhancing the enzyme’s antioxidant activity in the skin through structural stabilization and improved performance. The favorable binding interactions observed for gallic acid, phlorizin, and naringin further support their potential role in modulating GPx activity, thereby strengthening the skin’s defense mechanisms against oxidative stress. The effectiveness of these interactions is largely attributed to non-covalent forces, such as hydrogen bonding and aromatic interactions, which may enhance the bioavailability and efficacy of these compounds. Collectively, these findings highlight the potential of these natural antioxidants in dermatological formulations aimed at promoting skin health and protecting against oxidative damage. Although molecular docking results provide valuable information about potential interactions between the identified compounds and GPx, these results remain predictive in nature. Computational docking provides a useful preliminary understanding of binding affinity; however, it cannot fully reproduce the complexity of the enzymatic or cellular environment. Therefore, further experimental validation using enzymatic and cellular in vitro assays is necessary to confirm the biological significance of the predicted interactions.

### 2.8. Colloidal Analysis

Given the growing interest in plant-based cosmetic ingredients, hydrosols from *Artemisia* species have been evaluated for their physicochemical and colloidal properties relevant to topical formulations. As a result, plant hydrolates have become widely utilized as key ingredients in these natural cosmetics. Hydrolates, derived from plant extracts, are gaining popularity in the cosmetics industry and are now commonly found in many commercial products, presenting an interesting avenue for further research [22]. Unlike essential oils, hydrolates possess milder aromas, and their content of biologically active substances, along with their unique aromatic properties, makes them valuable ingredients in cosmetic formulations. Due to their high content of polar compounds, hydrolates are frequently used as the aqueous phase in emulsion-based products [23].

To support the use of plant hydrolates in cosmetics from a scientific perspective, it is essential to study their colloidal and chemical properties. The colloidal characteristics of the extracts play a crucial role in their cosmetic applicability. Reduced surface tension facilitates the even spreading of formulations on the skin surface, improving absorption and user comfort. Enhanced emulsion stability ensures uniform dispersion of bioactive compounds, which contributes to consistent biological activity and longer shelf-life of the product. Together, these properties translate into better formulation performance and increased efficacy in dermaceutical applications. For practical applications in cosmetic detergents and cleansers, it is particularly important to evaluate their ability to spread effectively. To this end, the extreme angles of transmission of aqueous hydrolate solutions onto a hydrophobic surface, such as Teflon (polytetrafluoroethylene), were measured. The adhesion isotherms presented in Figure 6A demonstrate that all hydrolates exhibit good hydrophilicity, even at low concentrations.

The cosθ = f(σ) dependencies indicate that complete hydrophilization of the Teflon surface is achieved. Surface tension measurements of aqueous solutions are crucial, as a decrease in surface tension provides a thermodynamic factor that contributes to the stability of dispersed systems, including cosmetic products. Accordingly, the surface tension of aqueous solutions of *A. albida* (AA), *A. leucodes* (AL), and *A. scopaeformis* (AS) hydrolates at concentrations of 0.5, 1.0, 1.5, and 2.0% (wt.) was measured at the liquid–gas interface. The resulting surface tension vs. concentration (σ = f(c)) isotherms, presented in Figure 6B, were obtained.

The isotherms reveal a typical trend as the concentration of the hydrolates increases, the surface tension decreases, indicating the presence of surfactant-like compounds. Both *A. albida* (AA) and *A. leucodes* (AL) hydrolates exhibit significant surface activity, reducing the surface tension of water to 40–50 mJ/m^2^, while *A. scopaeformis* (AS) shows only a slight effect on surface tension. Enhancing surface activity may be achieved by incorporating surfactants and examining their influence on the surface properties of hydrolates.

The pH values of the hydrolates, shown in Table 6, indicate that *A. leucodes* (AL) and *A. scopaeformis* (AS) are slightly acidic, while *A. albida* (AA) is neutral. These pH characteristics make the hydrolates suitable for use in a wide range of cosmetic formulations.

In this study, the potential for formulating emulsions based on *Artemisia* hydrolates, which are commonly used in cosmetic products such as creams, oils, and hydrophilic oils, was investigated [24,25]. The experiments revealed that the most stable emulsions were achieved with a volume ratio of sunflower oil to water of 6:4. Kinetic separation curves for sunflower oil/hydrolate (6:4) emulsions containing *A. albida* (AA), *A. leucodes* (AL), and *A. scopaeformis* (AS) hydrolates are presented in Figure 6C; similar results were observed for other hydrolates studied.

All *Artemisia* hydrolates showed comparable emulsifying ability, with *A. leucodes* demonstrating the greatest emulsion stability (35 min). The hydrolates exhibited moderate surface activity, good adhesion, slightly acidic to neutral pH, and moisturizing properties, making them suitable for cosmetic formulations such as creams, gels, and hydrophilic oils. Containing water-soluble antioxidants like flavonoids and polyphenols, *Artemisia* hydrolates offer similar benefits to essential oils but in a gentler form, safe for sensitive skin. Based on these findings, a moisturizing cream and healing ointment were developed, with future studies planned to evaluate their safety and efficacy.

Although this study provides comprehensive comparative data on the phytochemical composition, antioxidant activity, and colloidal behavior of three *Artemisia* species, several limitations must be acknowledged. The study was limited to samples collected from a single geographic location and harvest year, which may not fully reflect interannual or environmental variability in metabolite profiles. This study included both organic extracts and hydrolates to gain a comprehensive understanding of the dermatological potential of various products derived from *Artemisia* species. Although the extracts are rich in phenolic and flavonoid components suitable for chromatographic quantification, the hydrolates primarily contain volatile and water-soluble compounds responsible for their soothing and aromatic properties. Due to the very low concentration of UV-absorbing compounds, HPLC analysis of the hydrosols revealed insufficient signal intensity for reliable quantification. However, their biological activity and colloidal properties were assessed to illustrate their complementary roles in dermatology. Furthermore, biological activity was mainly assessed using in vitro biochemical assays without cellular or in vivo validation. Safety and toxicity assessments are beyond the scope of this study, but are important to confirm the suitability of these extracts for dermoceutical or dietary applications. Future studies combining multi-season sampling, extensive metabolic profiling, cytotoxicity screening, and efficacy testing in relevant biological models will only confirm, improve and extend the translational relevance of these findings.

## 3. Materials and Methods

### 3.1. General Experimental Procedures

Analytical balances (RADWAG Wagi Elektroniczne, Radom, Poland), SNOL 67/350 dryer (AB UMEGA, Utena, Lithuania), muffle furnace (Smolenskoye SKTB SPU, Smolensk, Russia), and desiccator were used to analyze the raw materials quantitatively, including moisture and ash content. Identifying extractive substances required a reflux condenser, drier, and balances. To qualitatively analyze the bioactive compounds such as flavonoids, free organic acids, alkaloids, saponins, tannins, and coumarins, equipment such as a flask, reflux condenser, and analytical balances were used. The contents of flavonoids, saponins and coumarins were evaluated through colorimetric assay approach using a UV-5500 UV-Vis spectrophotometer (Shanghai Metash Instruments Co., Ltd., Shanghai, China) at various wavelengths. The burette is used for measuring the content of free organic acids and alkaloids through acid–base titration techniques. Polysaccharides were detected by drying the extract to a consistent bulk. The mineral composition of samples was determined using a Shimadzu 6200 series spectrometer (Shimadzu Corporation, Japan). The chromatographic detection of organic active substituents was performed on HPLC LC-40 (Shimadzu Corporation, Kyoto, Japan). The standards of phenolic compounds including gallic acid (Chemical Abstract Service (CAS) Registry Number: 149-91-7), catechin (CAS: 154-23-4), epicatechin (CAS: 490-46-0), naringin (CAS: 480-41-1) and phlorizin (CAS: 60-81-1) were purchased from Shanghai Standard Technology Co., Ltd., Shanghai, China. The SpectraMax M3 Multi-Mode Microplate Reader (Molecular Devices, San Jose, CA, USA) was used for antioxidant assays and UV spectral studies. For colloidal properties such as surface tension, pH of solution and transmission property, the crux tool, 781 pH/Ion Meter (Metrohm, Herisau, Switzerland), potentiometer, and goniometer were used. The cream preparation process required common chemical laboratorial dishes such as heat-resistant glass cup, glass stick for stirring and thermometer for temperature control.

### 3.2. Plant Materials and Preparation of Extracts

For the study, three species of *Artemisia* L. were collected from different regions of Kazakhstan during their flowering period in September 2023. *A. albida* was collected in the Almaty region, known for its warm climate and location at the foot of the Zailiyskiy Alatau. *A. leucodes* was collected in the South Kazakhstan region, characterized by a sharply continental climate with hot summers and cold winters. *A. scopaeformis* was collected in the East Kazakhstan region, where foothills and highlands soften the extreme climatic conditions of deserts and semi-deserts.

Each plant specimen was carefully harvested during the peak growing season of September 2023, and the identification of the species was confirmed through morphological analysis in comparison with authenticated herbarium samples, and the voucher specimens of the plants (KAZNU-230901 (*A. albida*), KAZNU-230902 (*A. leucodes*), and KAZNU-230903 (*A. scopaeformis*)) were deposited at the Research Center for Medicinal Plants of Al-Farabi Kazakh National University, Almaty, Kazakhstan. The collected plant materials were cleaned, air-dried under controlled conditions, ground and stored for subsequent analysis.

After exhaustive extraction with 96% ethanol, the concentrated crude extract was sequentially partitioned with ethyl acetate and n-butanol to obtain fractions of increasing polarity. These extracts were used for comparative evaluation of phytochemical composition and biological activity. Hydrolates were collected as aqueous distillates obtained during the steam distillation of the aerial parts.

### 3.3. The Qualitative Analysis of Artemisia L. Species

The prominent bioactive components such as organic acids, flavonoids, polysaccharides, tannins, alkaloids, and coumarins of *Artemisia* spp. were quantitatively and qualitatively identified using methods described in the monographs [26,27]. The qualitative investigation on the composition of the selected bioactive constituents, as well as contents of moisture, ash and extractives (by 96% ethanol) of *Artemisia* species, was analyzed based on the procedure stated in the State Pharmacopeia of the Republic of Kazakhstan methodology [27].

### 3.4. Preparation of Artemisia L. Hydrolates

Hydrolates of *Artemisia* spp. were prepared in the laboratory of the Research Center for Medicinal Plants, Almaty, Kazakhstan, using a classical steam distillation method, widely regarded as the most effective technique for isolating co-distilled aqueous extracts suitable for therapeutic and cosmetic applications.

In this method, plant material was placed on a perforated grid above the boiling water level in a standard steam distillation apparatus. The water in the boiling flask was gradually heated to generate steam, which passed directly through the plant material. As the steam penetrated the plant tissues, it ruptured secretory structures such as oil glands, facilitating the release of volatile constituents. The vapor mixture, containing both water vapor and volatile plant compounds, was routed through a condenser maintained at a controlled temperature to ensure efficient condensation.

The resulting distillate was collected in a separatory vessel, where it spontaneously separated into two phases due to differences in polarity and density. The upper layer, consisting of essential oil, was removed manually, while the lower aqueous layer—the hydrolate—was collected, filtered, and stored at 4 °C in sterile place for further analysis.

### 3.5. Semi-Quantitative Analysis Substances by High-Performance Liquid Chromatography (HPLC)

The phenolic profile of *Artemisia* spp. extracts was determined using high-performance liquid chromatography (HPLC) on a Shimadzu LC-40 system, employing five standards of phenolic compounds that are gallic acid, catechin, epicatechin, naringin, phlorizin, rutin, and quercetin. All HPLC measurements were performed in triplicate, and the results are presented as mean ± standard deviation. In total, 1 mg of the dried extract was weighed and dissolved in 1 mL of methanol. An injection volume of 10 µL was introduced into the system, with the autosampler temperature maintained at 40 °C. The selected chemical constituents were separated on a reverse-phase C18 column (250 mm × 4.6 mm i.d., 5 µm particle size). The mobile phase consisted of solvent A (1% acetic acid in water, *v*/*v*) and solvent B (acetonitrile), delivered at a constant flow rate of 1.0 mL/min. A gradient elution was employed, beginning with 10% solvent B and linearly increasing to 90% over 55 min. The detection wavelength of 272 nm was selected after preliminary UV spectral scanning of the extracts, which showed strong and consistent absorption for all target compounds across different metabolite classes. This wavelength provided optimal overall sensitivity and resolution, allowing for the simultaneous detection of phenolic acids, flavonoids, and coumarins. Chromatographic operation, data acquisition, and analysis were carried out using Shimadzu LabSolutions V 5.1 software, which also facilitated peak integration and retention time determination. Determination of phenolic compounds was achieved by comparing their retention times to those of reference standards under identical chromatographic conditions, and quantification was performed with each individual compound calibration curves [28].

### 3.6. DPPH Scavenging Activity

The free radical scavenging activity of *Artemisia* spp. hydrolates was evaluated using the 2,2-diphenyl-1-picrylhydrazyl (DPPH) assay, following an established method with modifications adapted for high-throughput analysis in 96-well microplates [29]. In each well, 10 µL of the hydrolate sample was mixed with 190 µL of a 0.25 mM ethanolic DPPH solution. The reaction mixtures were incubated in the dark at 20 °C for 15 min. Following incubation, absorbance was measured at 517 nm using a microplate reader. The antioxidant activity is expressed as a percentage of DPPH radical inhibition, calculated using Equation (1). The half-maximal inhibitory concentration (IC_50_) was determined from the resulting dose–response curves.(1)$$Radical\;scavenging\;activity\;\left(\%\right)=\frac{A_{0}-A}{A}\times 100$$

### 3.7. Ferric Reducing Antioxidant Potential (FRAP) Assay

The ferric reducing antioxidant potential (FRAP) of *Artemisia* spp. extracts was determined using a published method with slight modifications [30]. This assay is based on the reduction of the colorless ferric complex (Fe^3+^-tripyridyltriazine) to the blue-colored ferrous complex (Fe^2+^-tripyridyltriazine) by the action of electron-donating antioxidants at low acidity. The reduction was monitored by measuring the change in absorbance at 593 nm. The working FRAP reagent was prepared by mixing 152 µL of 300 mM acetate buffer (pH 3.6), 19 µL of 10 mM TPTZ (2,4,6-tri(2-pyridyl)-s-triazine) containing 40 mM HCl, and 19 µL of 20 mM ferric chloride. In total, 10 μL of the *Artemisia* spp. hydrolates (1 mg/mL) was added to 190 µL of the prepared FRAP reagent. The difference in absorbance between the sample and the blank was measured and used to calculate the FRAP values.

### 3.8. Sun Protection Factor (SPF) Evaluation of Artemisia Species

The photoprotection capacity of *Artemisia* hydrolates was calculated by establishing the SPF value corresponding to the adapted process from a previous publication [29]. For this, the extracts were diluted to 11 different concentrations and their spectra were measured in the range from 290 to 320 nm with an interval of 5 nm. The *SPF* was then calculated using the equation below:(2)$$SPF=CF\;\times \;\sum_{290}^{320}EE\;\left(\lambda \right)\times I\;\left(\lambda \right)\times abs\;(\lambda )$$
where *CF* = correction factor which was calculated based on sunscreen with known *SPF* (10), *EE* (*λ*) = erythmogenic effect of radiation with wavelength *λ*, *Abs* (*λ*) = spectrophotometric absorbance values at wavelength *λ*. The values of *EE* × *I* are constant. These values are established in the previous published report [31,32].

### 3.9. DNA Oxidation Protection of Artemisia Extracts

The hydroxyl radical-induced DNA damage assay was conducted according to the method described by Baiseitova and co-researchers (2021) [21]. A reaction mixture was prepared by combining 1 µL of pBR322 plasmid DNA (0.35 µg/mL) in 9 µL of phosphate buffer (pH 7.4), 2 µL of 1 mM FeSO_4_, 5 µL of *Artemisia* spp. hydrolates, and 3 µL of 30% H_2_O_2_ in an Eppendorf tube. The mixture was incubated at 37 °C for 36 min in the dark. Following incubation, 5 µL of the reaction mixture was combined with 1 µL of 6× DNA loading buffer (Enzynomics, Daejeon, Republic of Korea) and loaded onto a 0.8% agarose gel containing RedSafe DNA dye (Intron Biotechnology, Seongnam, Republic of Korea) in TAE buffer (40 mM Tris-acetate, 1 mM EDTA). Electrophoresis was performed at 85 V for 30 min. DNA bands were visualized and recorded using Image Lab Software 6.0. The inhibition of DNA damage (%) was calculated using Equation (3).DNA damage (%) = ocDNA band intencity/pBR322 DNA band intensity × 100(3)

### 3.10. Molecular Docking Studies

Docking simulations of the chemical marker compounds identified by HPLC were conducted using Maestro 13.4 (Schrödinger, New York, NY, USA). The structure of glutathione peroxidase (GPx) was obtained from the Protein Data Bank (PDB) at http://www.rcsb.org accessed on 15 March 2025. The chemical structures of the compounds, which were previously identified as the chemical marker compounds through HPLC analysis, were generated using ChemDraw 19.0.0.22 software. These chemical marker compounds, serving as reference standards, were imported into Maestro 13.4 for further analysis. The 3D models of the compounds were optimized at pH 7.0 ± 2.0 using the LigPrep tool. Protein structure minimization was performed using the OPLS4 force field, iteratively refining the protein structure until the average root-mean-square deviation (RMSD) of the heavy atoms reached 0.3 Å. The PDB ID of the target protein (7U4N) was then incorporated. The receptor’s active site docking grid was generated by defining a receptor grid, with the active site centered in the grid box at coordinates X = 28.57, Y = −11.54, and Z = −9.53.

### 3.11. Colloidal Analysis of Artemisia Hydrolates

Hydrolate solutions AA, AL, and AS, consisting of *A. albida*, *A. leucodes*, and *A. scopaeformis*, respectively, were prepared at concentrations of 0.5%, 1.0%, 1.5%, and 2.0% (*w*/*v*). Surface tension was measured using the Du Noüy method, and the pH of the solutions was determined with a 781 pH/Ion Meter potentiometer. Transmission properties were assessed by measuring the critical angle of transmission using a goniometer. The emulsifying and foaming properties of the hydrolates were studied using an established method [33].

## 4. Conclusions

In this study, a comparative study of *A. albida*, *A. leucodes* and *A. scopaeformis* was conducted for the first time, focusing on their phytochemical composition, antioxidant potential, photoprotective properties and colloidal properties. All three species were found to be rich in bioactive compounds and essential elements, which confirms their potential use in cosmeceutical and nutraceutical formulations. *A. albida* was found to have the highest content of extractive substances (20.76%) and water-soluble polysaccharides (2.14%), indicating strong wound healing and immunomodulatory potential. Tannins were most abundant in *A. scopaeformis* (2.81%) and *A. albida* (1.52%), while coumarins were predominant in *A. scopaeformis* (6.49%) and *A. leucodes* (4.46%). Mineral analysis revealed significant Ca, K and Mg content in all species, with the highest concentrations observed in *A. scopaeformis* (Ca: 6.61 mg/g, K: 10.91 mg/g, Mg: 2.06 mg/g). Antioxidant studies showed that *A. leucodes* had the highest antioxidant activity, with an IC_50_ value for DPPH of 13.53 μM, a FRAP value of 52.02 μmol TE/g, an SPF of 23.24, and a DNA protection efficiency of 91.4%. These results indicate a high potential for use as a UV protection and anti-aging agent. Molecular docking confirmed the high binding of catechin and epicatechin to glutathione peroxidase, indicating a possible enhancement of the antioxidant enzyme function. In addition, *A. albida* and *A. leucodes* hydrolates demonstrated significant colloidal activity, reducing the surface tension to 40–50 mJ/m^2^, confirming their use as natural emulsifiers in topical formulations. Based on comprehensive phytochemical, biological and physical data, these species stand out as promising plants for the development of multifunctional skin care ingredients, sunscreens and antioxidant-enriched products. Overall, this study demonstrates the high dermatological and nutraceutical potential of these *Artemisia* extracts, supported by their rich phytochemical composition, antioxidant activity, and beneficial colloidal properties. These results provide a foundation for the development of natural formulas with protective and restorative effects on the skin. Further enzymatic and cellular studies are needed to confirm these results and determine directions for further product development.

## Figures and Tables

**Figure 1 molecules-30-04165-f001:**
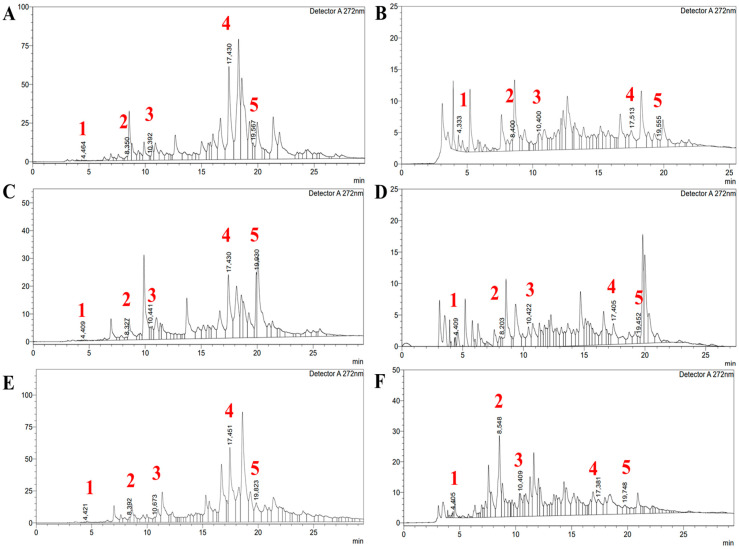
The HPLC spectra of *Artemisia* species: (**A**,**C**,**E**) Ethyl acetate and (**B**,**D**,**F**) Butanol fractions of *A. albida*, *A. leucodes*, and *A. scopaeformis*, respectively. Detected compounds are gallic acid (1), catechin (2), epicatechin (3), naringin (4), and phlorizin (5).

**Figure 2 molecules-30-04165-f002:**
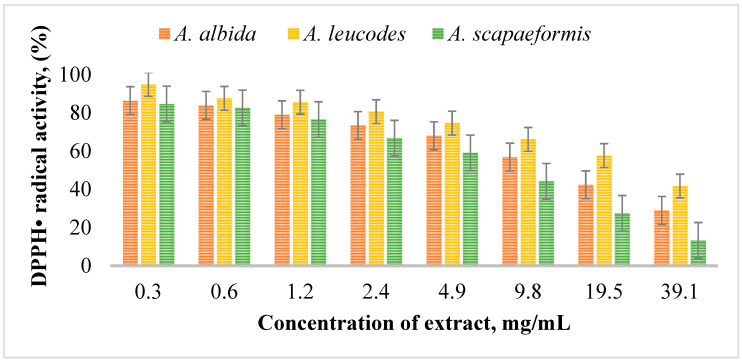
DPPH radical scavenging activity of *Artemisia* extracts at concentrations of 0.3~39.1 mg/mL.

**Figure 3 molecules-30-04165-f003:**
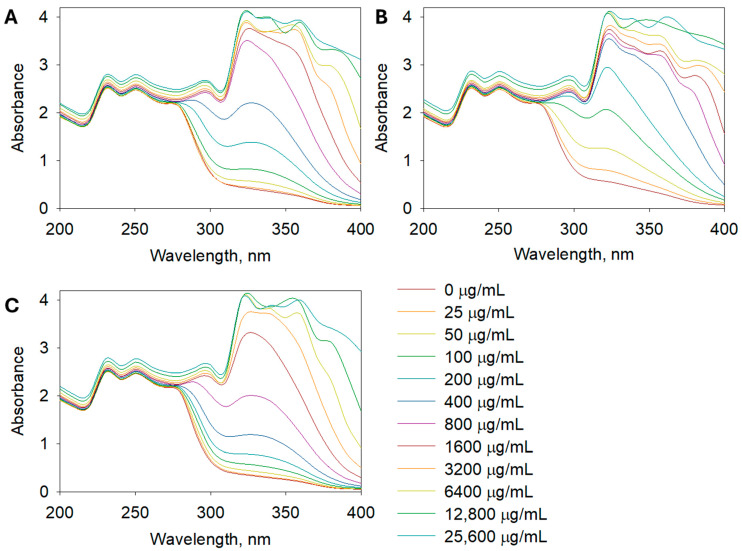
UV spectra of (**A**) *A. albida*, (**B**) *A. leucodes*, and (**C**) *A. scopaeformis* extracts at concentrations of 25~25,600 µg/mL.

**Figure 4 molecules-30-04165-f004:**
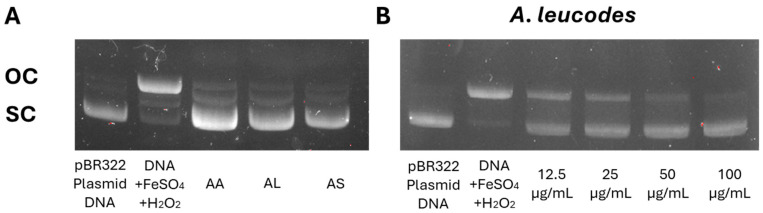
(**A**) Protective effect of *Artemisia* extracts at 100 μM and (**B**) Dose-dependent protective effect of *A. leucodes* against DNA damage induced by Fenton’s reaction.

**Figure 5 molecules-30-04165-f005:**
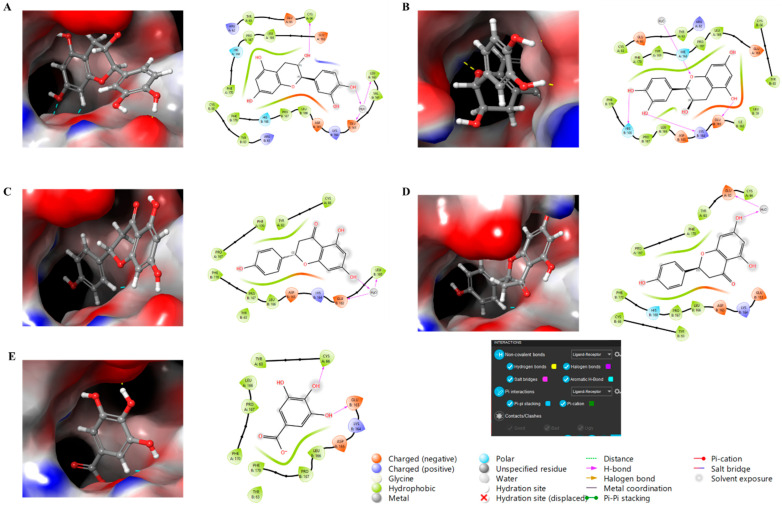
Molecular docking study of compounds against glutathione peroxidase with 3D and 2D representation. (**A**) Catechin, (**B**) Epicatechin, (**C**) Naringin, (**D**) Phlorizin, (**E**) Gallic Acid.

**Figure 6 molecules-30-04165-f006:**
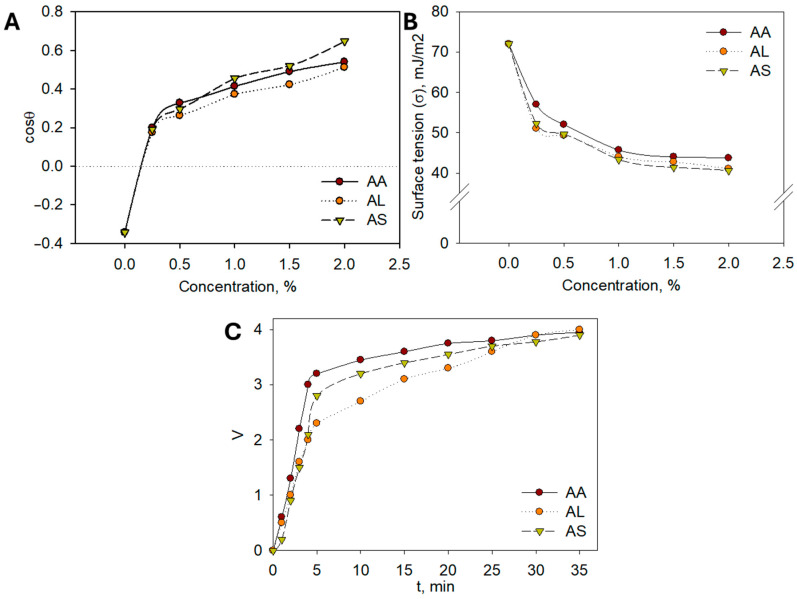
(**A**) Dilution isotherms of aqueous solutions of *Artemisia* hydrolates; (**B**) Surface tension isotherms of aqueous solutions of *Artemisia* hydrolates; (**C**) Kinetic stability curves of emulsions.

**Table 1 molecules-30-04165-t001:** Quantitative analysis of bioactive constituents of *Artemisia* L. species.

Component	Content, %		
	*A. albida*	*A. leucodes*	*A. scopaeformis*
Extractive substances	20.80 ± 0.08	14.40 ± 0.04	18.40 ± 0.07
Organic acids	0.21 ± 0.01	0.07 ± 0.01	0.18 ± 0.01
Flavonoids	0.03 ± 0.01	0.05 ± 0.01	0.18 ± 0.01
Polysaccharides	2.14 ± 0.02	2.00 ± 0.02	1.08 ± 0.01
Tannins	1.52 ± 0.02	0.88 ± 0.02	2.81 ± 0.03
Alkaloids	0.53 ± 0.01	0.75 ± 0.02	0.78 ± 0.01
Coumarins	1.60 ± 0.02	4.46 ± 0.05	6.49 ± 0.03

**Table 2 molecules-30-04165-t002:** Mineral composition of *Artemisia* L. species.

Element	Concentration in Ash, mg/g			Concentration in Plant, mg/g		
	*AA*	*AL*	*AS*	*AA*	*AL*	*AS*
Microelements						
Cd	0.002 ± 0.001	0.007 ± 0.001	0.005 ± 0.001	0.0001 ± 0.0001	0.0005 ± 0.0001	0.0003 ± 0.0001
Ni	0.039 ± 0.004	0.051 ± 0.002	0.038 ± 0.001	0.0024 ± 0.0002	0.0037 ± 0.0001	0.0025 ± 0.0001
Pb	n/d *	0.004 ± 0.001	n/d	n/d	0.0003 ± 0.0001	n/d
Cu	0.037 ± 0.003	0.064 ± 0.001	0.079 ± 0.001	0.0023 ± 0.0001	0.0047 ± 0.0002	0.0051 ± 0.0002
Mn	0.413 ± 0.005	0.451 ± 0.002	0.504 ± 0.003	0.0257 ± 0.0008	0.0331 ± 0.0008	0.0329 ± 0.0006
Zn	0.099 ± 0.005	0.108 ± 0.003	0.135 ± 0.002	0.0062 ± 0.0002	0.0079 ± 0.0004	0.0088 ± 0.0005
Fe	2.574 ± 0.005	4.557 ± 0.007	2.461 ± 0.003	0.1601 ± 0.0006	0.3345 ± 0.0002	0.1608 ± 0.0007
Macroelements						
Ca	46.08 ± 0.04	68.07 ± 0.01	101.20 ± 0.01	2.866 ± 0.002	4.996 ± 0.006	6.613 ± 0.009
Mg	16.86 ± 0.02	20.30 ± 0.01	31.50 ± 0.04	1.049 ± 0.001	1.490 ± 0.001	2.058 ± 0.002
Na	4.73 ± 0.02	4.96 ± 0.01	5.27 ± 0.03	0.294 ± 0.001	0.364 ± 0.001	0.344 ± 0.001
K	57.87 ± 0.04	89.87 ± 0.04	167.10 ± 0.03	3.600 ± 0.002	6.597 ± 0.009	10.910 ± 0.008

* n/d—not detected.

**Table 3 molecules-30-04165-t003:** Concentration of compounds analyzed by HPLC and the limit of detection (LOD) and quantification (LOQ).

Compounds	t_R_ (min)	Regression Equation	*r* ^2^	Mean of Slopes (S)	St. Dev. (σ)	LOD	LOQ
Gallic acid	4.4 ± 0.2 *	y = 62,908x − 4220.7	0.9993	60,210	59,753	3.275	9.924
Catechin	8.7 ± 0.4	y = 13,380x + 13,886	0.9995	13,309	3577	0.887	2.688
Epicatechin	10.4 ± 0.4	y = 16,766x + 17,149	0.9995	16,699	3985	0.787	2.386
Naringin	17.4 ± 0.7	y = 29,474x + 31,639	0.9994	29,445	2161	0.242	0.734
Phlorizin	19.6 ± 0.5	y = 24,892x + 28,318	0.9994	24,844	7623	1.012	3.068

* All the tested compounds were examined in triplicates.

**Table 4 molecules-30-04165-t004:** The standard phenolics content of *Artemisia* species analyzed by HPLC.

*Artemisia* Species	*A. albida*		*A. leucodes*		*A. scopaeformis*	
Fractions	EA	BuOH *	EA *	BuOH	EA	BuOH
Compounds	Concentration, μg/1 g of dry mass of the plant					
Gallic acid	0.88 ± 0.03	3.41 ± 0.11	0.93 ± 0.02	2.49 ± 0.22	0.90 ± 0.02	2.36 ± 0.10
Catechin	15.74 ± 0.54	5.21 ± 0.49	1.35 ± 0.80	2.10 ± 0.77	28.59 ± 0.74	271.21 ± 2.46
Epicatechin	2.40 ± 1.64	3.72 ± 0.04	15.80 ± 1.21	26.64 ± 0.87	17.74 ± 1.78	45.73 ± 1.00
Naringin	354.32 ± 1.15	11.48 ± 0.04	129.55 ± 0.07	12.78 ± 0.02	386.73 ± 10.57	38.30 ± 002
Phlorizin	491.90 ± 3.60	0.30 ± 0.32	64.43 ± 3.42	1.65 ± 0.85	164.49 ± 2.0	12.00 ± 0.66

* EA—ethyl acetate, BuOH—butanol.

**Table 5 molecules-30-04165-t005:** Summary of antioxidant in radical scavenging experiments.

Extracts	DPPH IC_50_ (μM)	FRAP ^a^ (μmol TE/g)	SPF ^b^	DNA Damage Protection ^c^ (%)
*A. albida*	20.04 ± 1.27	43.91 ± 2.31	14.59 ± 1.03	86.3
*A. leucodes*	13.53 ± 0.71	52.02 ± 3.61	23.24 ± 0.71	91.4
*A. scopaeformis*	28.87 ± 0.85	32.37 ± 1.22	9.93 ± 0.92	82.1
Trolox ^d^	10.21 ± 1.06	-	-	NT ^e^

^a^ The concentrations are the FRAP values; ^b^ SPF value of extracts at concentration of 200 μg/mL; ^c^ DNA damage protection were calculated for 100 μg/mL of extracts; ^d^ Trolox was used as a positive control; ^e^ NT is not tested.

**Table 6 molecules-30-04165-t006:** Colloidal–chemical properties of *Artemisia* hydrolates.

Hydrolate	Adhesion Work, mJ/m^2^	Adhesion Work W_max_, mJ/m^2^	pH	Emulsion “Life” Time, min
AA	28.52 ± 0.04	67.16 ± 0.01	6.5	20
AL	24.10 ± 0.02	72.56 ± 0.05	6.2	35
AS	21.16 ± 0.02	74.58 ± 0.04	6.7	22

## Data Availability

The original contributions presented in this study are included in the article. Further inquiries can be directed to the corresponding authors.

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
