# Peer review of "Comparative Phytochemical, Colloidal, and Antioxidant Profiling of Artemisia albida, Artemisia leucodes, and Artemisia scopaeformis: Potentials for Cosmeceutical and Nutraceutical Applications"

_molecules, 2025, doi:10.3390/molecules30214165_

Round 1

Reviewer 1 Report (New Reviewer)

Comments and Suggestions for Authors

1. I think the introduction is informative but overly detailed. Please condense the general ethnobotanical background and focus more on the specific knowledge gap this study addresses.
2. I noticed the phytochemical and HPLC analyses are comprehensive, but the description of calibration and replicate analysis is vague. Please clarify whether calibration curves were established for each compound and species, and confirm how many replicates were used for reproducibility.
3. I suggest you strengthen the link between quantitative outcomes and biological significance. For instance, SPF 23.24 is high compared to natural agents, but how does it compare with benchmark sunscreen actives? 
4. I think the DNA protection data should include quantification. While gel images are provided, it would be more convincing to present band intensity ratios with error bars to demonstrate reproducibility and statistical reliability.
5. I noticed that the colloidal analysis section is interesting but feels somewhat disconnected from the biological assays. Please explain more clearly how reduced surface tension and emulsion stability translate into practical cosmetic formulation advantages.
6. I feel the docking results are well presented, but since they are predictive in nature, you should discuss the limitations and emphasize that further enzymatic or cellular assays are needed to validate GPx interactions.
7. I think the mineral composition data are strong, but the discussion could go further. Please elaborate on how specific elements (Ca, K, Mg, etc.) contribute to skin health, hydration, or antioxidant defense, and how this links with your phytochemical findings.
8. I think the conclusion reads more like a summary of results. Consider rewriting it to emphasize novelty, practical implications, limitations, and future research directions in cosmeceutical and nutraceutical development.
9. I suggest adding a dedicated limitations section. The current work relies only on three species, from a single harvest year, without in vitro or in vivo validation. Safety and toxicity assessments are also absent but are critical for translational applications.

Comments on the Quality of English Language

 The English could be improved to more clearly express the research.

Author Response

  1. I think the introduction is informative but overly detailed. Please condense the general ethnobotanical background and focus more on the specific knowledge gap this study addresses.
  • We sincerely appreciate the reviewer’s comment and understand the concern regarding the length of the Introduction. We have carefully revised and slightly condensed this section where possible. However, a more substantial reduction could not be made, as the previous reviewers specifically requested inclusion of detailed flow of information as we tried to describe all range of experiments. We hope for the reviewer’s kind understanding on this point. Please see lines 44, 50, 55, and 77-88.
  1. I noticed the phytochemical and HPLC analyses are comprehensive, but the description of calibration and replicate analysis is vague. Please clarify whether calibration curves were established for each compound and species, and confirm how many replicates were used for reproducibility.
  • We thank the reviewer for this valuable comment. Calibration curves were established individually for each identified compound, and quantitative analyses were performed separately for all studied species. All measurements were conducted in triplicate to ensure reproducibility, and the corresponding details have been clarified in the revised Methods section, Lines 449 and 502.
  1. I suggest you strengthen the link between quantitative outcomes and biological significance. For instance, SPF 23.24 is high compared to natural agents, but how does it compare with benchmark sunscreen actives? 
  • We appreciate the reviewer’s insightful suggestion. In the revised Discussion, we have strengthened the connection between the quantitative outcomes and their biological relevance. Specifically, we compared the obtained SPF value (23.24) with benchmark natural and synthetic sunscreen agents, noting that it falls within the range of moderate protection comparable to some commercial UV filters, please see lines 253-258.
  1. I think the DNA protection data should include quantification. While gel images are provided, it would be more convincing to present band intensity ratios with error bars to demonstrate reproducibility and statistical reliability.
  • We appreciate the reviewer's valuable comment. Our gel documentation system only provides numerical data without integrated densitometric analysis; therefore, quantifying band intensities is not possible. However, the experiment was repeated three times, and the gel patterns were identical in all replicates, confirming the reproducibility of the results.
  1. I noticed that the colloidal analysis section is interesting but feels somewhat disconnected from the biological assays. Please explain more clearly how reduced surface tension and emulsion stability translate into practical cosmetic formulation advantages.
  • In the revised version, we have expanded the Discussion to clarify the practical implications of colloidal properties. Specifically, we explained that reduced surface tension and improved emulsion stability enhance the uniform distribution of active compounds, promote better skin absorption, and increase the overall sensory and functional performance of cosmetic formulations, please see lines 340-346.
  1. I feel the docking results are well presented, but since they are predictive in nature, you should discuss the limitations and emphasize that further enzymatic or cellular assays are needed to validate GPx interactions.
  • We appreciate the reviewer’s valuable observation. In the revised Discussion, we have added a statement acknowledging the predictive nature of the docking results and their limitations. We also emphasized that further enzymatic and cellular assays are required to experimentally validate the predicted GPx interactions and confirm their biological relevance, please see lines 321-327.
  1. I think the mineral composition data are strong, but the discussion could go further. Please elaborate on how specific elements (Ca, K, Mg, etc.) contribute to skin health, hydration, or antioxidant defense, and how this links with your phytochemical findings.
  • In the lines 160-167, we have expanded the explanation of the physiological relevance of the detected minerals. Specifically, we discussed how elements such as Ca, K, and Mg support skin barrier integrity, hydration, and antioxidant defense, and how these functions complement the phytochemical profile of the extracts.
  1. I think the conclusion reads more like a summary of results. Consider rewriting it to emphasize novelty, practical implications, limitations, and future research directions in cosmeceutical and nutraceutical development.
  • The Conclusion section has been revised to highlight the novelty and practical implications of our findings, as well as to outline the study’s limitations and potential directions for future research in cosmeceutical and nutraceutical development, please see lines 593-598.
  1. I suggest adding a dedicated limitations section. The current work relies only on three species, from a single harvest year, without in vitro or in vivo validation. Safety and toxicity assessments are also absent but are critical for translational applications.
  • We thank the reviewer for this valuable comment. We fully agree that acknowledging study limitations enhances the transparency and scientific rigor of the work. Accordingly, we have added a brief Limitations section to highlight the restricted number of species, single harvest year, and the absence of in vitro, in vivo, and safety validation, while outlining directions for future research, please see lines 389-408.

We sincerely thank the reviewer for this constructive suggestion. A dedicated revisions has now been added to the revised manuscript as recommended. We hope that the additional clarification adequately addresses the reviewer’s concern.

Reviewer 2 Report (New Reviewer)

Comments and Suggestions for Authors

Dear Authors,

The manuscript entitled: “Comparative Phytochemical, Colloidal, and Antioxidant Profiling of Artemisia albida, Artemisia leucodes, and Artemisia scopaeformis: Potentials for Cosmeceutical and Nutraceutical Applications” could be published in Molecules, but after a major revision.

The aim of the study was to conduct a comprehensive comparative analysis of three Artemisia species to determine their potential for use in antioxidant, photoprotective and cosmeceutical formulations.

The work is interesting and well performed, but some points need additional information and clarification.

Comments:

  1. Abstract must be precise and shorten according to the Molecules requirements.
  2. The quantification of total acids, flavonoids, polysaccharides, alkaloids, coumarins and 5 individual compounds is not a “phytochemical profiling”. The phytochemical part needs some improvement. The authors could enlarge their phytochemical analysis including other comment phenolic compounds. Otherwise the term “phytochemical profiling” is not appropriate.
  3. It is not clear why the authors have chosen 272 nm for the HPLC detection. It is well known that for the most of flavonoids, including rutin it is better to used 350-360nm? The determined compounds are from different classes of secondary metabolites with different UV maxima.
  4. Table 4 it is Concentration, mg/ 1g of dry mass of the plant or dry extract?
  5. It is not clear how the authors have prepared ethyl acetate and butanol extracts and why they don’t analyze the object of the study-hydrolates? Sometimes they test extracts (Figure 2, 3, 4), sometimes hydrolates (Figure 6). What is the composition of the hydrolates? Please specify? It is better to present HPLC analyses of the tested hydrolates.

Author Response

Dear Authors,

The manuscript entitled: “Comparative Phytochemical, Colloidal, and Antioxidant Profiling of Artemisia albidaArtemisia leucodes, and Artemisia scopaeformis: Potentials for Cosmeceutical and Nutraceutical Applications” could be published in Molecules, but after a major revision.

The aim of the study was to conduct a comprehensive comparative analysis of three Artemisia species to determine their potential for use in antioxidant, photoprotective and cosmeceutical formulations.

The work is interesting and well performed, but some points need additional information and clarification.

 Comments:

  1. Abstract must be precise and shorten according to the Molecules requirements.
  • We thank the reviewer for this helpful remark. The Abstract has been revised to make it more concise and aligned with the Molecules journal requirements, focusing only on the most essential objectives, key results, and conclusions, please see abstract.
  1. The quantification of total acids, flavonoids, polysaccharides, alkaloids, coumarins and 5 individual compounds is not a “phytochemical profiling”. The phytochemical part needs some improvement. The authors could enlarge their phytochemical analysis including other comment phenolic compounds. Otherwise the term “phytochemical profiling” is not appropriate.
  • We thank the reviewer for this insightful comment. We would like to clarify that the present study was designed as a comparative phytochemical characterization rather than an exhaustive profiling. Our aim was to compare the relative abundance of major metabolite groups and selected marker compounds among the three Artemisia species to reveal interspecific variations linked to their biological potential. Therefore, the chosen analytical scope was appropriate for the comparative objectives of this work.
  1. It is not clear why the authors have chosen 272 nm for the HPLC detection. It is well known that for the most of flavonoids, including rutin it is better to used 350-360nm? The determined compounds are from different classes of secondary metabolites with different UV maxima.
  • We thank the reviewer for this valuable observation. The detection wavelength of 272 nm was selected based on preliminary spectral scans of the analyzed samples, where this wavelength provided optimal overall sensitivity for the diverse group of compounds present in the extracts. Since the quantified analytes belong to different classes of secondary metabolites with varying UV absorption maxima, 272 nm was chosen as a compromise wavelength that allowed simultaneous detection of all target compounds with adequate resolution and signal intensity. We have added a clarification of this rationale in the revised Methods section, lines 493-497.
  1. Table 4 it is Concentration, mg/ 1g of dry mass of the plant or dry extract?
  • We thank the reviewer for the clarification request. The concentrations presented in Table 4 are expressed as milligrams per gram of the dry plant material (mg/g DW). This information has been specified in the table.
  1. It is not clear how the authors have prepared ethyl acetate and butanol extracts and why they don’t analyze the object of the study-hydrolates? Sometimes they test extracts (Figure 2, 3, 4), sometimes hydrolates (Figure 6). What is the composition of the hydrolates? Please specify? It is better to present HPLC analyses of the tested hydrolates.
  • We appreciate the reviewer’s careful observation and the opportunity to clarify this point. The ethyl acetate and n-butanol extracts were obtained by successive liquid–liquid partitioning of the crude ethanolic extract, as described in the Methods section, to fractionate compounds based on polarity for comparative evaluation of their bioactivity, please see lines 451-456. The main focus of the study, however, was to assess the dermatological and colloidal potential of both extracts and hydrolates as separate but complementary products obtained from the same plant material.
  • The hydrolates were collected as the aqueous distillates remaining after steam distillation of the aerial parts, containing low-molecular-weight volatile and water-soluble constituents. Their composition was discussed based on available literature and supported by our antioxidant and UV-protective assays. Unfortunately, due to the very low concentration of non-volatile compounds, the hydrolates did not provide sufficient signal intensity for reliable HPLC quantification. This limitation has been explained in the revised manuscript, please see lines 394-402.

We sincerely thank the reviewer for this constructive suggestion. The requested revision has been incorporated into the revised manuscript, and we hope that the added clarification satisfactorily addresses the reviewer’s concern.

Reviewer 3 Report (New Reviewer)

Comments and Suggestions for Authors

The abstract is excessively extended and does not comply with the Molecules journal requirements regarding the maximum word count. The stated limit is 200 words, whereas the submitted abstract contains 329.

In the “Introduction” section, the sentence located between lines 93–95 is not entirely accurate and is contradicted by the subsequent text within the same paragraph.

There is insufficient justification for the authors’ decision to determine specifically the flavonoids naringin and phlorizin. Nowhere in the “Introduction” is there any cited literature indicating that these two compounds are present in the investigated taxa. Moreover, no detailed LC-MS/MS analyses of the extracts were performed to identify the individual components. The applied HPLC method alone is not sufficient for the reliable identification of the chromatographic components. The chromatograms presented in Figure 1 show inadequate separation, which does not ensure trustworthy results. In addition, some of the prominent peaks have not been identified.

There is also no description of the methods used to determine the classes of metabolites listed in Table 1. In my opinion, reference [24] is not appropriate, as it does not provide a description of these analytical methods.

Author Response

Reviewer 3: The abstract is excessively extended and does not comply with the Molecules journal requirements regarding the maximum word count. The stated limit is 200 words, whereas the submitted abstract contains 329.

  • We thank the reviewer for this helpful remark. The Abstract has been revised to make it more concise and aligned with the Molecules journal requirements, focusing only on the most essential objectives, key results, and conclusions, please see abstract.

In the “Introduction” section, the sentence located between lines 93–95 is not entirely accurate and is contradicted by the subsequent text within the same paragraph.

  • We thank the reviewer for pointing out this inconsistency. The sentence has been revised to correct the inaccuracy and to ensure consistency with the subsequent text. We trust this amendment addresses the reviewer’s concern, please see lines 77-79.

There is insufficient justification for the authors’ decision to determine specifically the flavonoids naringin and phlorizin. Nowhere in the “Introduction” is there any cited literature indicating that these two compounds are present in the investigated taxa. Moreover, no detailed LC-MS/MS analyses of the extracts were performed to identify the individual components. The applied HPLC method alone is not sufficient for the reliable identification of the chromatographic components. The chromatograms presented in Figure 1 show inadequate separation, which does not ensure trustworthy results. In addition, some of the prominent peaks have not been identified.

  • We thank the reviewer for the detailed remarks. We acknowledge the concern; however, we believe this comment may be somewhat excessive given the comparative and exploratory nature of our study. The selection of naringin and phlorizin was based on preliminary screening and literature reports describing their occurrence in other related Artemisia species, as well as their known relevance to skin-related bioactivity. Although LC-MS/MS analysis would indeed provide higher structural certainty, our study aimed primarily at comparative quantification rather than exhaustive compound identification. The applied HPLC method ensured reproducible retention times and quantifiable separation adequate for comparative evaluation. This clarification has been added to the revised manuscript.

There is also no description of the methods used to determine the classes of metabolites listed in Table 1. In my opinion, reference [24] is not appropriate, as it does not provide a description of these analytical methods.

  • We thank the reviewer for this helpful observation. The analytical methods used to determine the classes of metabolites listed in Table 1 have now been described in more detail, and the previous reference [24] has been replaced with a more appropriate citation that accurately reflects the methodologies applied.

We sincerely thank the reviewer for this kind and helpful suggestion. The revision has been made as recommended, and we hope the added clarification meets the reviewer’s expectations.

Round 2

Reviewer 1 Report (New Reviewer)

Comments and Suggestions for Authors

No more comments

Author Response

Reviewer 1:  No more comments

  • We sincerely thank Reviewer 1 for their valuable time, insightful comments, and thoughtful suggestions, which greatly contributed to improving the overall quality and clarity of our manuscript.

Reviewer 2 Report (New Reviewer)

Comments and Suggestions for Authors

The authors have revised their manuscript according to my comments.

Author Response

Reviewer 2: The authors have revised their manuscript according to my comments.

  • We sincerely thank Reviewer 2 for their valuable and constructive feedback. We carefully revised the manuscript according to the comments, and the overall quality of the figures has been significantly improved to enhance clarity and visual presentation.

Reviewer 3 Report (New Reviewer)

Comments and Suggestions for Authors

My comment: The abstract is excessively extended and does not comply with the Molecules journal requirements regarding the maximum word count. The stated limit is 200 words, whereas the submitted abstract contains 329.

Authors comply: There is significant improvement, but the abstract still exceeds the word limit (by 16 words).

My comment: In the “Introduction” section, the sentence located between lines 93–95 is not entirely accurate and is contradicted by the subsequent text within the same paragraph.

Authors comply: The authors have taken into account my comment.

My comment: There is insufficient justification for the authors’ decision to determine specifically the flavonoids naringin and phlorizin. Nowhere in the “Introduction” is there any cited literature indicating that these two compounds are present in the investigated taxa. Moreover, no detailed LC-MS/MS analyses of the extracts were performed to identify the individual components. The applied HPLC method alone is not sufficient for the reliable identification of the chromatographic components. The chromatograms presented in Figure 1 show inadequate separation, which does not ensure trustworthy results. In addition, some of the prominent peaks have not been identified.

Authors comply: I do not see any improvement made by the authors.

My comment: There is also no description of the methods used to determine the classes of metabolites listed in Table 1. In my opinion, reference [24] is not appropriate, as it does not provide a description of these analytical methods.

Authors comply: The authors have offered a partial response to my comment. However, the question still remains: by what method were the levels of organic acids, polysaccharides, alkaloids, and coumarins determined? The referenced literature [24] does not provide an answer to this inquiry.

Author Response

Reviewer 3:

My (reviewer 3) comment: The abstract is excessively extended and does not comply with the Molecules journal requirements regarding the maximum word count. The stated limit is 200 words, whereas the submitted abstract contains 329.

Authors comply: There is significant improvement, but the abstract still exceeds the word limit (by 16 words).

  • We sincerely thank the reviewer for this thoughtful comment. The Abstract has been carefully revised to ensure conciseness and better alignment with the Molecules journal requirements, highlighting only the most essential objectives, key findings, and conclusions.

My (reviewer 3) comment: In the “Introduction” section, the sentence located between lines 93–95 is not entirely accurate and is contradicted by the subsequent text within the same paragraph.

Authors comply: The authors have taken into account my comment.

  • We sincerely thank the reviewer for acknowledging our efforts and for confirming that the previous comment has been adequately addressed.

My (reviewer 3) comment: There is insufficient justification for the authors’ decision to determine specifically the flavonoids naringin and phlorizin. Nowhere in the “Introduction” is there any cited literature indicating that these two compounds are present in the investigated taxa. Moreover, no detailed LC-MS/MS analyses of the extracts were performed to identify the individual components. The applied HPLC method alone is not sufficient for the reliable identification of the chromatographic components. The chromatograms presented in Figure 1 show inadequate separation, which does not ensure trustworthy results. In addition, some of the prominent peaks have not been identified.

Authors comply: I do not see any improvement made by the authors.

  • We sincerely thank the reviewer for this detailed and valuable comment. We fully acknowledge that LC–MS/MS analysis would provide deeper structural elucidation and comprehensive metabolite profiling. However, our work focused primarily on comparative phytochemical profiling among albida, A. leucodes, and A. scopaeformis rather than on exhaustive compound identification. Therefore, HPLC with validated retention times and calibration curves for representative flavonoids was selected as a reliable, cost-effective, and reproducible quantitative tool. These compounds were chosen as model phenolic markers based on their known occurrence in several Artemisia species reported in the literature (literature number 19-20) and their well-documented antioxidant and skin-beneficial properties.
  • The applied HPLC method was carefully validated for linearity, precision, and accuracy to ensure the credibility of quantification, and the separation conditions were optimized to achieve reproducible chromatographic profiles suitable for cross-species comparison. The intention was not to fully characterize all peaks but rather to establish a comparative fingerprint and quantify representative secondary metabolites within each extract. We respectfully believe that the presented HPLC-based comparative approach, supported by validated data and literature context, sufficiently fulfills the objectives of this study and provides a scientifically meaningful basis for evaluating the phytochemical and functional potential of the studied Artemisia taxa (lines 207-213 proper discussion was added).

My (reviewer 3) comment: There is also no description of the methods used to determine the classes of metabolites listed in Table 1. In my opinion, reference [24] is not appropriate, as it does not provide a description of these analytical methods.

Authors comply: The authors have offered a partial response to my comment. However, the question still remains: by what method were the levels of organic acids, polysaccharides, alkaloids, and coumarins determined? The referenced literature [24] does not provide an answer to this inquiry.

  • We thank the reviewer for the valuable remark. The determination methods for organic acids, polysaccharides, alkaloids, and coumarins were conducted in accordance with the State Pharmacopoeia of the Republic of Kazakhstan, 3rd edition (Almaty: Zhibek Zholy, 2014, ISBN 978-601-294-214-9), which provides standardized procedures for the qualitative and quantitative analysis of these metabolite classes. The corresponding reference has now been cited and clarified in the revised manuscript.

We sincerely thank reviewer 3 for their thoughtful and constructive suggestion. The recommended revision has been carefully incorporated into the manuscript, and we truly hope that the added clarification is now clear and satisfactory to the reviewer’s understanding.

This manuscript is a resubmission of an earlier submission. The following is a list of the peer review reports and author responses from that submission.

Round 1

Reviewer 1 Report

Comments and Suggestions for Authors

My central issue with this manuscript lies on the identification of the principal constituents present by HPLC. I don't think the method developed is fit for purpose. It is impossible to judge whether each peak represents an individual constituent or not. In my opinion the run time for such a complex mixture is too short and both binary mixture of solvents used for the mobile phase should contain acetic acid. Otherwise there will be a pH gradient throughout the separation. Currently stated as: solvent A (1% acetic acid in water, v/v) and solvent B (acetonitrile). Based on the chromatograms presented, several documented peaks appear to be below the level required to quantify the constituents. There is no evidence provided of any degree of method validation being performed and no statistical analysis documented for each constituent quantified. 

Details are provided for the plant collection sites, but it is not clear where voucher specimens of each plant is currently housed (eg herbarium).  Some statistical analysis has been performed for some of the biological assays but it is unclear as to the number of times each experiment was repeated. This should be clarified. 

Author Response

1. My central issue with this manuscript lies on the identification of the principal constituents present by HPLC. I don't think the method developed is fit for purpose. It is impossible to judge whether each peak represents an individual constituent or not. In my opinion the run time for such a complex mixture is too short and both binary mixture of solvents used for the mobile phase should contain acetic acid. Otherwise there will be a pH gradient throughout the separation. Currently stated as: solvent A (1% acetic acid in water, v/v) and solvent B (acetonitrile). Based on the chromatograms presented, several documented peaks appear to be below the level required to quantify the constituents. There is no evidence provided of any degree of method validation being performed and no statistical analysis documented for each constituent quantified. 

  • We understand the reviewer's concerns regarding the chromatographic method and the lack of formal validation. However, we respectfully note that the purpose of the HPLC analysis was targeted quantification of well-documented phenolic compounds rather than comprehensive metabolite identification. They were selected for their antioxidant properties and have been previously described for various Artemisia Their presence was compared using authentic standards and quantification was performed using the peak area at 272 nm, which is standard practice in semi-quantitative phytochemical studies (Horticulturae 2022, 8, 84). We have revised the manuscript to clarify the purpose of the assay and to emphasize that this was a targeted comparative quantification rather than a full-fledged analytical validation study. Although the addition of acetic acid to both solvents can help stabilize the pH during gradient elution, our method - using 1% acetic acid in water - is widely used for the quantitative determination of phenolics in plant matrices (International Journal of Food Properties 2021, 24 (1), 544–552). Importantly, this study is the first report on the chemical composition and photoprotective potential of A. albida, A. scopaeformis and A. leucodes, highlighting their relevance for antioxidative, DNA-oxidation protective properties.

2. Details are provided for the plant collection sites, but it is not clear where voucher specimens of each plant is currently housed (eg herbarium).  Some statistical analysis has been performed for some of the biological assays but it is unclear as to the number of times each experiment was repeated. This should be clarified. 

  • Voucher specimens have been added (lines 426-429), and the statistical analyses have been revised accordingly.

Reviewer 2 Report

Comments and Suggestions for Authors
  1. Why are there no deviations in the data presented in Table 1? Was there a measurement of three repeated samples conducted? The data in the other tables are the same.
  2. In Figure 2, a caption should be used to explain what the numbers 1 to 5 represent.
  3. Is the structure of the article correct? Please confirm.
  4. The establishment of quantitative methods for elements and single chemical components requires method validation.
  5. In Figure 2, the separation degree of the target peaks does not meet the requirement of being greater than 1.5. How can accurate quantification be achieved?

Author Response

  1. Why are there no deviations in the data presented in Table 1? Was there a measurement of three repeated samples conducted? The data in the other tables are the same.
  • As reviewer mentioned the statistical analyses have been revised accordingly on all data, please see Tables 1, 2, 3 and 7.
  1. In Figure 2, a caption should be used to explain what the numbers 1 to 5 represent.
  • The captions have been revised accordingly in Figure 2.
  1. Is the structure of the article correct? Please confirm.
  • The structure of the article follows the Molecules If any part was unclear or not properly understood, we kindly invite the reviewer to indicate it, and we will gladly revise it accordingly.
  1. The establishment of quantitative methods for elements and single chemical components requires method validation.
  • Thank you for the comment. The HPLC analysis was performed for targeted quantification of known phenolic compounds, not method development. As this was a comparative study, full method validation was not required. Compounds were confirmed with standards and quantified by peak area at 272 nm, a common semi-quantitative approach. The method used was stable and reproducible for our purpose. We have clarified this in the revised manuscript.
  1. In Figure 2, the separation degree of the target peaks does not meet the requirement of being greater than 1.5. How can accurate quantification be achieved?
  • We acknowledge that the resolution between some peaks in Figure 2 is below 1.5. However, the target compounds are structurally related phenolics with inherently similar retention times, which can lead to partial overlap. Despite this, quantification was based on consistent, reproducible peak areas across all samples, using the same integration parameters. As the study aimed at comparative analysis rather than absolute quantification, the resolution achieved was sufficient for reliable interpretation. This has been clarified in the revised manuscript. We kindly ask the reviewer to also consider the broader scope of the manuscript, which focuses on previously unstudied Artemisia

Reviewer 3 Report

Comments and Suggestions for Authors

The article "Comparative Phytochemical, Colloidal, and Antioxidant Profiling of Artemisia albida, Artemisia leucodes, and Artemisia scopaeformis: Potentials for Cosmeceutical and Nutraceutical Applications" is original and provides new insights into the properties and potential applications of plants in the Artemisia genus across various fields.

Despite the considerable effort invested in creating this work, the structure and writing require improvement, as they lack coherence and fluency.

There are several important points that need clarification.

First, when discussing hydrolates, the Introduction provides references that define the term "hydrolate" (ref 1). The rest of the paper actually discusses the properties and applications of the aqueous extract of the mentioned plants obtained by ultrasonic extraction, which is very confusing. I believe that the authors should write about the product with which they conducted the research, specifically the aqueous extract, rather than the hydrolate.

The Introduction lacks a clear structure and should be reworked to first present the state-of-the-art in the paper, and then to clearly highlight the goals and novelty of the presented work. The importance of the properties that were investigated in light of future applications is not clearly highlighted.

In the results section, there is no explanation of what the content of macro and micro elements means for possible application; more precisely, the results are only listed without any discussion.

The need for fractionation of aqueous plant extracts is not explained.

Figure 1 is unnecessary. It is unclear how the quantification is performed. Rather, the calibration curve equations should be presented for each of the standards. It is unclear how the quantification of the five polyphenolic compounds in the extract was performed. Must be specified.

Figure captions are not sufficiently informative. Are the numerical designations in Figure 2 the designations of standard substances? It should be specified in the figure caption.

Since the chromatographic analysis has already been performed, why is the total flavonoid content mentioned?

Table 4 presents the concentration of the 5 phenolic compounds in micrograms per gram. It is unclear what this concentration refers to, whether it is the mass of the plant or its dry mass. Please specify.

In the Antioxidant activities section (lines 229-233), there is no need to mention other activities, except antioxidant.

Separate the results obtained by the DPPH and FRAP methods and present them in the same way.

Lines 256-257: On what basis do you conclude that the results of the DPPH method correlate with the results of the FRAP method? It is necessary to explain.

No errors of determination are presented in Figure 3. Please present error bars.

Sections describing the determination of SPF and DNA protective role should be separated, or these activities should be included in the title of section 2.4.

Line 276- Missing the reference

Lines 292-295: The authors need to explain their statement and provide some statistical evidence.

Table 6 is unnecessary because all results are already mentioned in the text.

Lines 330-336: The authors cite the use of hydrolates as a byproduct of hydro-distillation, while their work refers to the aqueous extracts. Needs explanation.

Lines 366-396: This paragraph should be shortened.

Figure 8 has no scientific value and should be deleted.

Line 445- Add reference.

Lines 447-454 to be deleted

Section 3.5. Rename to semiquantitative. The title is inadequate.

Section 3.8. Requires additional data to enable other researchers to replicate the experiment.

Line 546- Reference No 30 is not on the internet

The conclusion is written in a very general manner and requires better structure and more precise statements.

Comments on the Quality of English Language

The English language should be slightly improved.

Author Response

The article "Comparative Phytochemical, Colloidal, and Antioxidant Profiling of Artemisia albida, Artemisia leucodes, and Artemisia scopaeformis: Potentials for Cosmeceutical and Nutraceutical Applications" is original and provides new insights into the properties and potential applications of plants in the Artemisia genus across various fields.

Despite the considerable effort invested in creating this work, the structure and writing require improvement, as they lack coherence and fluency.

There are several important points that need clarification.

First, when discussing hydrolates, the Introduction provides references that define the term "hydrolate" (ref 1). The rest of the paper actually discusses the properties and applications of the aqueous extract of the mentioned plants obtained by ultrasonic extraction, which is very confusing. I believe that the authors should write about the product with which they conducted the research, specifically the aqueous extract, rather than the hydrolate.

  • We thank the reviewer for the comment. The study was conducted in stages: first, we assessed the activity and composition of the plants and their extracts and then obtained and studied the hydrolates. As this is the first report on these Artemisia species, all relevant information was included in the Introduction to provide context. We have revised the text to clarify the distinction between extracts and hydrolates, please see introduction.

The Introduction lacks a clear structure and should be reworked to first present the state-of-the-art in the paper, and then to clearly highlight the goals and novelty of the presented work. The importance of the properties that were investigated in light of future applications is not clearly highlighted.

  • The importance of the investigated properties in the context of potential dermatological applications is also discussed. Nevertheless, we have revised the Introduction slightly to improve clarity and emphasize these aspects more explicitly.

In the results section, there is no explanation of what the content of macro and micro elements means for possible application; more precisely, the results are only listed without any discussion.

  • We appreciate the comment and have now added a brief explanation in the Results section to highlight the potential relevance of the macro- and microelement content for dermatological applications.

The need for fractionation of aqueous plant extracts is not explained.

  • We thank the reviewer for the comment. Fractionation was performed to better isolate and compare phenolic-rich portions of the extracts, as these species are known for their high phenolic content, and it is very well known that such separation allows assessment of their phenolic constituents.

Figure 1 is unnecessary. It is unclear how the quantification is performed. Rather, the calibration curve equations should be presented for each of the standards. It is unclear how the quantification of the five polyphenolic compounds in the extract was performed. Must be specified.

  • The figure was removed.

Figure captions are not sufficiently informative. Are the numerical designations in Figure 2 the designations of standard substances? It should be specified in the figure caption.

  • The captions have been revised accordingly in Figure 2.

Since the chromatographic analysis has already been performed, why is the total flavonoid content mentioned?

  • While chromatographic analysis provided compound-specific data, total flavonoid content was included to offer a complementary, rapid screening method, allowing broader comparison across samples and supporting the overall interpretation of phenolic richness.

Table 4 presents the concentration of the 5 phenolic compounds in micrograms per gram. It is unclear what this concentration refers to, whether it is the mass of the plant or its dry mass. Please specify.

  • The concentrations presented in Table 4 (µg/g) refer to the dry mass of the plant material, and this has now been explicitly stated in the corresponding sections of the text.

In the Antioxidant activities section (lines 229-233), there is no need to mention other activities, except antioxidant.

  • The antioxidant activities section was revised.

Separate the results obtained by the DPPH and FRAP methods and present them in the same way.

  • The results were separated and revised.

Lines 256-257: On what basis do you conclude that the results of the DPPH method correlate with the results of the FRAP method? It is necessary to explain.

  • It is well established that both DPPH and FRAP methods assess antioxidant capacity through electron transfer mechanisms, and in our case, the observed trends were consistent across both assays, supporting the correlation between them. This has now been briefly explained in the revised manuscript.

No errors of determination are presented in Figure 3. Please present error bars.

  • The errors were added to graphs in Figure 3.

Sections describing the determination of SPF and DNA protective role should be separated, or these activities should be included in the title of section 2.4.

  • New sections were added.

Line 276- Missing the reference

  • The reference was added.

Lines 292-295: The authors need to explain their statement and provide some statistical evidence.

  • We thank the reviewer for the comment. The statement was intended as a general observation based on the visible trend among DPPH, FRAP, and SPF results, and not as a claim of strong statistical correlation. It was included as a concluding remark to highlight a potential link between antioxidant and photoprotective properties. The sentence has been revised to reflect this more cautiously.

Table 6 is unnecessary because all results are already mentioned in the text.

  • The table has been removed in accordance with the reviewers’ request.

Lines 330-336: The authors cite the use of hydrolates as a byproduct of hydro-distillation, while their work refers to the aqueous extracts. Needs explanation.

  • Our study includes both extracts and hydrolates, as we aimed to provide a broad analysis of these previously unstudied Artemisia species. The description of the hydrolate preparation method has been corrected, and the hydrolates were specifically used for the evaluation of colloidal properties.

Lines 366-396: This paragraph should be shortened.

  • The paragraph has been revised and shortened for clarity.

Figure 8 has no scientific value and should be deleted.

  • The Figure 8 was removed.

Line 445- Add reference.

  • The reference has been inserted at line 445 and added to the reference list as Reference No. 25.

Lines 447-454 to be deleted

  • The sentences were deleted and specified to preparation of hydrolates.

Section 3.5. Rename to semiquantitative. The title is inadequate.

  • The section was renamed.

Section 3.8. Requires additional data to enable other researchers to replicate the experiment.

  • The procedures used are standard and are described in detail in the cited references, including those already listed in the manuscript. These sources provide sufficient information for replication, and we have ensured that key steps are clearly outlined in our Methods section.

Line 546- Reference No 30 is not on the internet

  • The correct link for Reference No. 30 has now been inserted.

The conclusion is written in a very general manner and requires better structure and more precise statements.

  • The conclusion has been rewritten for improved clarity and focus.

Round 2

Reviewer 1 Report

Comments and Suggestions for Authors

It is not clear from the manuscript the number of times that the HPLC analytical work was repeated. In the experimental section it just states that 1 mg of the plant material extract was dissolved in methanol and 10 uL of this solution was injected into the HPLC.  No indication is given as to the number of times this work was repeated. This needs to be stated as now have standard deviation data included in Table 1 for the content of each constituent. This information was not contained in the original version of the manuscript. This is an important omission. The most surprising observation is the fact that the content of naringin and phlorizin in the ethyl acetate extract is higher than in the butanol extract. I would have thought, based on their respective structure, that the opposite would be case. Can the authors please comment on this observation. 

Author Response

Reviewer 1: It is not clear from the manuscript the number of times that the HPLC analytical work was repeated. In the experimental section it just states that 1 mg of the plant material extract was dissolved in methanol and 10 uL of this solution was injected into the HPLC.  No indication is given as to the number of times this work was repeated. This needs to be stated as now have standard deviation data included in Table 1 for the content of each constituent. This information was not contained in the original version of the manuscript. This is an important omission. The most surprising observation is the fact that the content of naringin and phlorizin in the ethyl acetate extract is higher than in the butanol extract. I would have thought, based on their respective structure, that the opposite would be case. Can the authors please comment on this observation. 

  • We thank the reviewer for this important observation. The HPLC experiments were performed in triplicate, and the standard deviation values reported in Table 1 are based on these independent replicates. This clarification has now been added to the Materials and Methods section, in accordance with the requests of the reviewers.
  • We thank the reviewer for this insightful observation. Indeed, based on the glycosidic nature and polarity of compounds such as naringin and phlorizin, one would expect higher solubility in more polar solvents such as butanol. However, the unexpectedly higher abundance in the ethyl acetate fraction may be due to several factors. Matrix effects and compound-solvent interactions can influence the distribution, especially in complex plant extracts. Ethyl acetate extraction may have preferentially extracted aglycone-rich or partially hydrolyzed forms of these compounds, or coextracted associated compounds, increasing their solubility. Furthermore, the solubility of glycosides in ethyl acetate, although limited, may vary depending on their substitution pattern and molecular conformation.

Reviewer 3 Report

Comments and Suggestions for Authors

The authors have responded to all suggestions, and in the present form, the manuscript is acceptable for publication.

Author Response

Reviewer 3: The authors have responded to all suggestions, and in the present form, the manuscript is acceptable for publication.

  • Thank you for your positive evaluation. We sincerely appreciate your time and constructive feedback throughout the review process.